# *PSEN1* His214Asn Mutation in a Korean Patient with Familial EOAD and the Importance of Histidine–Tryptophan Interactions in TM-4 Stability

**DOI:** 10.3390/ijms25010116

**Published:** 2023-12-21

**Authors:** Eva Bagyinszky, Minju Kim, Young Ho Park, Seong Soo A. An, SangYun Kim

**Affiliations:** 1Department of Industrial and Environmental Engineering, Graduate School of Environment, Gachon University, Seongnam 13120, Republic of Korea; eva85@gachon.ac.kr; 2Department of Neurology, Seoul National University College of Medicine & Clinical Neuroscience Center, Seoul National University Bundang Hospital, Seongnam 13620, Republic of Korea; kmj682908@gmail.com (M.K.); kumimesy@snubh.org (Y.H.P.); 3Department of Bionano Technology, Gachon Medical Research Institute, Gachon University, Seongnam 13120, Republic of Korea

**Keywords:** *PSEN1*, early onset Alzheimer’s disease, mutation, whole-exome sequencing

## Abstract

A pathogenic mutation in presenilin-1 (*PSEN1*), His214Asn, was found in a male patient with memory decline at the age of 41 in Korea for the first time. The proband patient was associated with a positive family history from his father, paternal aunt, and paternal grandmother without genetic testing. He was diagnosed with early onset Alzheimer’s disease (EOAD). *PSEN1* His214Asn was initially reported in an Italian family, where the patient developed phenotypes similar to the current proband patient. Magnetic resonance imaging (MRI) scans revealed a mild hippocampal atrophy. The amyloid positron emission tomography (amyloid-PET) was positive, along with the positive test results of the increased amyloid ß (Aβ) oligomerization tendency with blood. The *PSEN1* His214 amino acid position plays a significant role in the gamma–secretase function, especially from three additional reported mutations in this residue: His214Asp, His214Tyr, and His214Arg. The structure prediction model revealed that PSEN1 protein His214 may interact with Trp215 of His-Trp cation-π interaction, and the mutations of His214 would destroy this interaction. The His-Trp cation-π interaction between His214 and Trp215 would play a crucial structural role in stabilizing the 4th transmembrane domain of PSEN1 protein, especially when aromatic residues were often reported in the membrane interface of the lipid–extracellular region of alpha helices or beta sheets. The His214Asn would alter the cleavage dynamics of gamma–secretase from the disappeared interactions between His214 and Trp215 inside of the helix, resulting in elevated amyloid production. Hence, the increased Aβ was reflected in the increased Aβ oligomerization tendency and the accumulations of Aβ in the brain from amyloid-PET, leading to EOAD.

## 1. Introduction

Early onset Alzheimer’s disease (EOAD) is a rarely occurring form of Alzheimer’s disease (AD), which usually appears between 30 and 65 years of age [1]. Besides memory decline, atypical disease phenotypes are reported, including motor dysfunctions (spastic paraparesis, Parkinsonism, and ataxia), language issues, and/or abnormal behavior [2,3]. Two main inclusions are identified in the brains of AD patients: the extracellular amyloid plaques and the intracellular neurofibrillary tangles. However, atypical inclusions may be possible, such as Pick’s bodies or Lewy bodies [3]. Genetics play a significant role in EOAD, with three major causative genes: amyloid precursor protein (*APP*, NC_000021.9), presenilin1 (*PSEN1*, NC_000014.9), and presenilin 2 (*PSEN2*, NC_000001.11). Among them, the majority of AD-related mutations are found in PSEN1, with more than 360 reported mutations and more than 100 mutations from Asian countries (http://www.alzforum.org/mutations/psen-1, accessed on 1 October 2023).

*PSEN1* protein is a component of the gamma–secretase complex (γ-sec) in processing APP, NOTCH3, and other transmembrane proteins through protease cleavages. The mutant *PSEN1* would alter the γ-sec related mechanisms by preventing the initial endoproteolytic (or epsilon, ε-cleavage) process of APP protein. This could result in elevated productions of long amyloid- (Aβ42,43) and reduced productions of short (Aβ40, Aβ38, and Aβ37) amyloid peptides. The longer amyloid peptides were associated with an increased ability to aggregate into senile plaques [4,5,6]. Furthermore, *PSEN1* mutations would also impact non-amyloid-related processes, including Notch signaling, calcium-related pathways, and/or beta-cadherin processes [1,3,4,5].

In this study, a pathogenic mutation in *PSEN1*, His214Asn, was found in a male EOAD patient in Korea for the first time. The proband patient developed memory dysfunctions and personality changes in his early 40s, with a positive family history of several similar symptoms. This mutation was initially reported in an Italian family with EOAD, and our case would be the first case of the *PSEN1* His214Asn mutation in Asia.

## 2. Methods

Whole blood was drawn from proband patient with blood collection tube of EDTA anticoagulant (3.0 mL BD Vacutainer^®^ plastic P700 plasma tube, Franklin Lakes, NJ, USA). White blood cells were separated from plasma after the centrifugation on 800× *g* for 30 min. The DNA was isolated from white blood cells with Qiagen Maxi blood kit (Seoul, Republic of Korea) by following the manufacturer’s protocol. Whole-exome sequencing (WES) analyses of DNA sequencing were performed using NextSeq 500 platform by Macrogen Inc. (https://dna.macrogen.com/, accessed on 1 October 2023, Seoul, Republic of Korea). Other causative mutations from other risk genes for neurodegenerative diseases, such as AD, Parkinson’s disease (PD), frontal temporal disease (FTD), amyotrophic lateral sclerosis (ALS), and vascular diseases, were investigated [7]. The sequencing data were checked by Integrative Genomics Viewer (IGV) tool (https://igv.org/doc/desktop/, accessed on 1 October 2023). The probable pathogenic mutation was also verified with Sanger sequencing. Primers were designed by Primer3Plus online tool (https://www.bioinformatics.nl/cgi-bin/primer3plus/primer3plus.cgi, accessed on 1 October 2023), and sequencing was performed by BioNeer company (https://www.bioneer.co.kr/, accessed on 1 October 2023 Dajeon, Republic of Korea). For the association analyses between the different mutation carrier genes, STRING version 12.0 (https://string-db.org/, accessed on 1 October 2023) and Cytoscape Cluego tools were used, version 3.10.0 (https://apps.cytoscape.org/apps/cluego, accessed on 1 October 2023 [7].

Pathogenicity prediction was performed on *PSEN1* His214Asn mutation using different tools, including PolyPhen-2 (http://genetics.bwh.harvard.edu/pph2/, accessed on 1 October 2023), SIFT (http://sift.jcvi.org/, accessed on 1 October 2023), and CADD version 1.6 (https://cadd.gs.washington.edu/, accessed on 1 October 2023) tools. The 3D protein structure predictions were performed on normal PSEN1 214 His and mutations in His214 (His214Asn, His214Asp, His214Tyr and His214Arg) using Phyre2 tool (http://www.sbg.bio.ic.ac.uk/phyre2/html/page.cgi?id=index, accessed on 1 October 2023). The structural alterations between *PSEN1* 214His and all pathogenic mutations in residue 214 were visualized by the Discovery Studio 3.5 Visualizer tool (https://discover.3ds.com/discovery-studio-visualizer-download, accessed on 1 October 2023).

Detailed clinical, imaging, and structure prediction data will be presented. This study was approved by the Institutional Review Board of Seoul National University Bundang Hospital (B-2311-867-703).

## 3. Clinical Features

The proband patient (III-1) was a 41-year-old man whose initial symptom was a short-term memory impairment and change in personality since the age of 38. His family history is notable due to three additional cases of progressive cognitive deterioration from three members of his family, with similar ages at onset and clinical phenotype. The father (II-1) of the proband patient had memory deficits in his early 50s and passed away at the age of 55. The paternal aunt (II-4) also experienced memory deficits in her early 50s and died in her mid-50s. Additionally, the paternal grandmother (I-2) of the proband exhibited memory deficits from her mid-50s and passed away in her early 60s. Genetic testing could not be performed on the above three members of the family (I-2, II-1, and II-4) due to their passing. All other living family members refused genetic testing or did not provide any further information regarding their health (Figure 1).

A previous neuropsychological assessment of the proband patient revealed deficits in attention, visuospatial function, short- and long-term memory, and executive functions. His clinical dementia rating (CDR) score was 0.5/3, with a sum of box (SOB) 2.5, and the mini mental status exam (MMSE) score was 27/30. The apolipoprotein E (*APOE*) genotype was E3/E3. The magnetic resonance image (MRI) indicated mild hippocampal atrophy and mild white matter hyperintensities (Figure 2A,B). There were no significant steno-occlusive lesions or aneurysms in the intracranial artery in magnetic resonance angiography (MRA). Amyloid positron emission tomography (amyloid-PET with ^18^F Florebetaben (FBB)-PET) showed amyloid positivity in several brain regions, including bilateral lateral temporal–frontal–parietal areas, posterior cingulate and precuneus (Figure 2C). During his initial visit to the hospital, the blood was drawn with heparin anti-coagulant and tested for amyloid ß (Aβ) oligomerization tendency with AlzOn (PeopleBio Inc., Sungnam, Republic of Korea) [8,9]. The AlzOn results were positive, with a value of 1.045 for the increased Aβ oligomerization tendency, confirming a high-risk profile. His treatment was started with Donepezil (10 mg/day). Two years later, the proband patient was retested with AlzOn, and the value was 1.009, indicating the continuous existence of Aβ progression and potential use of AlzOn in the clinical monitoring. A follow-up assessment two years later revealed a gradual progressive pattern of memory deficits, accompanied by an increase in depressive mood. His CDR, SOB, and MMSE scores were 1/3, 6, and 22/30, respectively, supporting gradual declines.

## 4. Results

Whole-exome sequencing analysis revealed a *PSEN1* His214Asn (g.chr14,73192735,C/A; c.640C>A; p.H214N) mutation in Figure 3. Sanger sequencing verified the presence of the mutation. PolyPhen2 predicted the *PSEN1* His214Asn mutation as damaging, with a score of 1. Multiple sequence alignment revealed that His214 would be a conserved amino acid residue among vertebrates since all available species (such as mice, bovines, chicken, or pufferfish) contained a histidine amino acid in the same position as their respective presenilin proteins. SIFT scoring also predicted the mutation as damaging, with a score of 0.01. The CADD tool also revealed high scores (26.6), suggesting a mutation with significant disturbances in structure and functions.

Structure prediction revealed that mutation would result in disturbances in the helix of the PSEN1 protein. *PSEN1* His214 was located on the border of the transmembrane helix (TM)-4 and hydrophilic loop (HL)-4. Our model revealed that His214 formed a hydrogen bond with Met210 and Ile211 and a cation-π interaction with Trp215. In the case of the Asn214 mutant, the H-bonds with Met210 and Ile211 remained, but the hydrophobic interaction with Trp215 was lost. In the case of His214Asn, the loss of hydrophobic interaction of the imidazole ring of histidine and Trp215 would cause the abnormal motion of helix-4. Since histidine and Asparagine are a positively charged aromatic and a neutral polar amino acid, respectively, His>Asn mutation would cause extra stress in the helix of the PSEN1 protein of intermolecular interactions with the lipid bilayer of the plasma membrane and its putative binding partners (Figure 4). Structure predictions were performed on other mutations in the PSEN1 His214 residue compared to the normal PSEN1 protein (Appendix A). His214Asp (Appendix A) and His214Arg (Appendix A) resulted in similar changes in intramolecular interactions to His214Asn since these mutations also destroyed the contact between His214 and Trp215. However, in the case of His214Tyr (Appendix A), the interactions remained between Try214 and Trp215 between both aromatic rings of histidine and tyrosine. However, the contact between His214-Trp215 and Tyr214-Trp215 would be altered due to the different properties of their aromatic side chain of histidine and tyrosine residues. Histidine is a positively charged residue with an imidazole side chain, while tyrosine is an uncharged residue with a hydrophobic hydroxyl-phenyl side chain.

Several common and rare variants were found in the patient in terms of AD, PD, and other neurodegenerative genes (Appendix A). In terms of other AD risk genes, rare variants were found in GRB2 associated binding protein 2 (*GAB2*; p.Gly135Val), Zinc Finger CW-Type And PWWP Domain Containing 1 (*ZCWPW1*; p.Ala210Pro), Translocase Of Outer Mitochondrial Membrane 40 (*TOMM40*; p.Leu312Val), NME/NM23 Family Member 8 (*NME8*; p.Pro147Leu), Ras And Rab Interactor 3 (*RIN3*; p.Gly972del), and Desmoglein 2 (*DSG2*; p.Ala662Thr) genes. Additional variants in PD risk genes included Serine/Threonine Kinase 39 (*STK39*; p.Gly221Ser) and Leucine Rich Repeat Kinase 2 (*LRRK2*; p.Asn1286Ser and p.His2391Gln). Furthermore, rare variants in other neurodegenerative disease-associated genes were Notch Receptor 3 (*NOTCH3*; p.Val237Met and DEAD-Box Helicase 1 (*DDX1*; p.Phe540Leu). STRING analysis (Figure 5) on these rare variants-carrier genes suggested the strong correlations of PSEN1 with *LRRK2*, *NOTCH3*, *ZCWPW1*, and *NME8* and indirect interactions with *RIN3* (through *ZCWPW1* and *NME8*). Interestingly, ClueGo did not find strong associations between the rare variant carrier genes.

## 5. Discussion

A pathogenic *PSEN1* His214Asn mutation was found in a 46-year-old Korean male EOAD patient with a likely strong family history of dementia. *PSEN1* His214Asn was a previously reported pathogenic mutation from an Italian patient female patient in 2016 (Table 1) [10]. Our case was the first discovered case of an Asian patient. The Italian patient developed memory dysfunctions at the age of 46, while the symptoms of memory impairments started earlier in the Korean patient (late 30s and early 40s). In terms of phenotypes, the Korean and Italian patients developed similar clinical symptoms, such as dysfunctions in short- and long-term memories and abnormal executive functions. In the Italian patient with *PSEN1* His214Asn, the disease duration was relatively long, and her condition worsened in the following 8 years from the appearance of her first symptoms. During her disease course, additional symptoms appeared, including aphasia, apraxia, agnosia, and seizure. In the final disease stages, the patient developed acute gastroenteric syndrome and kidney failure, and she fully lost her ability to speak. She also had akinetic mutism, drooling, severe tetraparesis, and dystonia, and later, she died from pneumonia. The computed tomography (CT) scan revealed diffuse cerebral atrophy, which was more prominent in the medial temporal lobes. Abnormalities were detected in electroencephalography (EEG), including disorganized background activity and abnormal theta–delta waves. However, no amyloid-PET or any biomarker data were performed/uncovered on the Italian patient. The Korean patient showed hippocampal atrophy in MRI, and PET revealed amyloid positivity in several brain regions. Also, his plasma MDS showed positive results for Aβ oligomers. In terms of family history, the Italian patient seemed to be positive, since her grandfather and mother also developed memory dysfunctions and dementia in their 50s [10]. The Korean patient also had affected family members, since his grandmother, father, one of his aunts, and his brother developed cognitive dysfunctions. Since family members refused the genetic test in both cases, segregation could not be determined in either the Italian or Korean patient.

Besides His214Asn, three additional probable pathogenic mutations were reported at the histidine 214 residue: His214Asp, His214Arg, and His214Tyr (Table 1). All of them were suggested to be the cause of the development of EOAD. His214Asp was discovered in a patient with atypical AD, who had the symptoms of motor impairment, such as bradykinesia and tremor, at the age of 55. The family history of the patient was probably positive, since her father and grandmother developed late-onset dementia. However, no additional details were available on detailed clinical symptoms, imaging, or biomarker status [11]. Next, the His214Tyr mutation was discovered in French and Iranian families. The French family revealed a strong family history of dementia among several members who were diagnosed with EOAD in their late 30s or early 40s. No further details were mentioned regarding the clinical symptoms, imaging data, or biomarker data [12]. The second case of His214Tyr was reported in an Iranian patient, who developed the first clinical symptoms (apathy and depression) at the age of 51, followed by progressive memory impairment, aphasia, and myoclonic jerks. In the later stage of the disease, the dementia progressed to more severe stages, including the appearance of hallucinations. The biomarker analysis for 14-3-3 protein was negative. Family history seemed to be positive, since at least seven family members were affected with variable symptoms, including motor and language impairments, and were diagnosed either with AD, Parkinson’s disease, or frontotemporal dementia. Segregation analysis could not be performed [13].

*PSEN1* His214Arg was reported in a Chinese patient who experienced dementia and behavioral issues in her 40s. Her MRI showed mild demyelination in the white matter in the frontal and parietal lobes; it did not detect any atrophy in the cerebral cortex or hippocampus. Family history seemed to be positive, and affected family members experienced memory dysfunctions, mood swings, and personality changes in their 40s [14].

Functional studies were performed on His214Asn, His214Asp, and His214Tyr. These three mutations were transfected into HEK293 cells by Liu et al. (2023). All of them showed an elevated ratio of Aβ42/Aβ40 and a lower ratio of Ab37/Aβ42, suggesting that these mutations would reduce the cleavage of long amyloid peptides, such as Aβ42, unexpectedly [15].

The *PSEN1* His214 position seemed to be an important residue in *PSEN1*, especially with four pathogenic mutations and confirmations of increased amyloid production in cell studies. Some researchers (2019) suggested that His214 would play a crucial role in APP binding and forming the substrate-binding pore of γ-sec [16]. *PSEN1* His214Asn was located in the fourth transmembrane helix domain and closer to the fourth hydrophilic loop (Figure 6). Our structure prediction revealed that His214 would have cation-π interaction with Trp215, but not in the case of Asn214, Asp214, or Arg214. Interestingly, the interaction between Tyr214 and Trp215 remained; however, the different properties of histidine and tyrosine residues may result in disturbances inside the contact, such as altered helix dynamics. Aromatic and charged interactions would often take a structural anchor inside the protein. The contact between histidine and tryptophan in the protein would play an important role in helix stability [17]. In the gas phase, histidine forms cation-π interactions with aromatic amino acids (such as Trp, Phe, or Tyr), and among them, the interaction between histidine and tryptophan was suggested to be the strongest. The energy of cation-π interaction between histidine and tryptophan could be -13.6 kcal/mole, which was higher than the energy between histidine and phenylalanine or histidine between tyrosine (−7.809 and −7.887 kcal/mol, respectively). Inside the proteins, histidine may form a π-π stacking interaction between the other aromatic residues nearby. The π-π stacking energy was also higher between histidine and tryptophan (−4.035 kcal/mol) compared to His-Tyr and His-Phe interactions (−3.463 and −3.084 kcal/mol, respectively) [18]. The aromatic residues, including Trp, may be located in the interface of the lipid–extracellular region of alpha helices or beta sheets. Trp would play a significant role in the stabilization of membrane proteins and protein folding, and its interactions with other aromatic residues would be essential for protein stability [19,20]. These findings suggested that the contacts between His214 and Trp215 in the helix-loop border would be crucial in helix stability and dynamics. The loss of this cation-π interaction would result in abnormal structural dynamics of the fourth transmembrane domain, leading to abnormal γ-sec functions [18].

## 6. Conclusions

In conclusion, a pathogenic *PSEN1* His214Asn mutation was discovered from a Korean EOAD patient for the first time in Asia. The age of onset and clinical symptoms of the Korean proband patient was similar to the first reported Italian EOAD patient with same mutation. These two cases occurred independently without any geographical or family connections. A limitation of this study was the lack of participation in genetic analyses for determining segregation results among family members with or without symptoms. The cation-π interaction between His214 and Trp215 would be crucial for the maintenance of the helix for proper PSEN1 function. All mutations at the His214 position would result in abnormal PSEN1 dynamics in the γ-secretase complex, leading to elevated and long amyloid production.

## Figures and Tables

**Figure 1 ijms-25-00116-f001:**
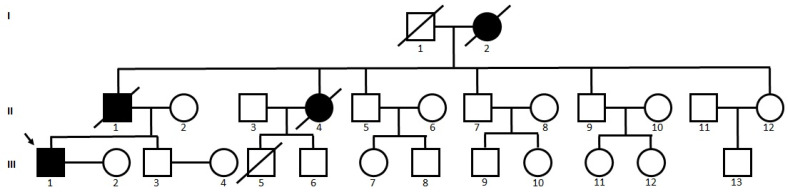
Family tree of proband patient with *PSEN1* His214Asn (III-1). Three additional family members (I-2, II-1 and II-4) also developed memory dysfunctions, but segregation could not be proven. Black colors are the family members, affected with memory decline. Black arrow shows the proband patient.

**Figure 2 ijms-25-00116-f002:**
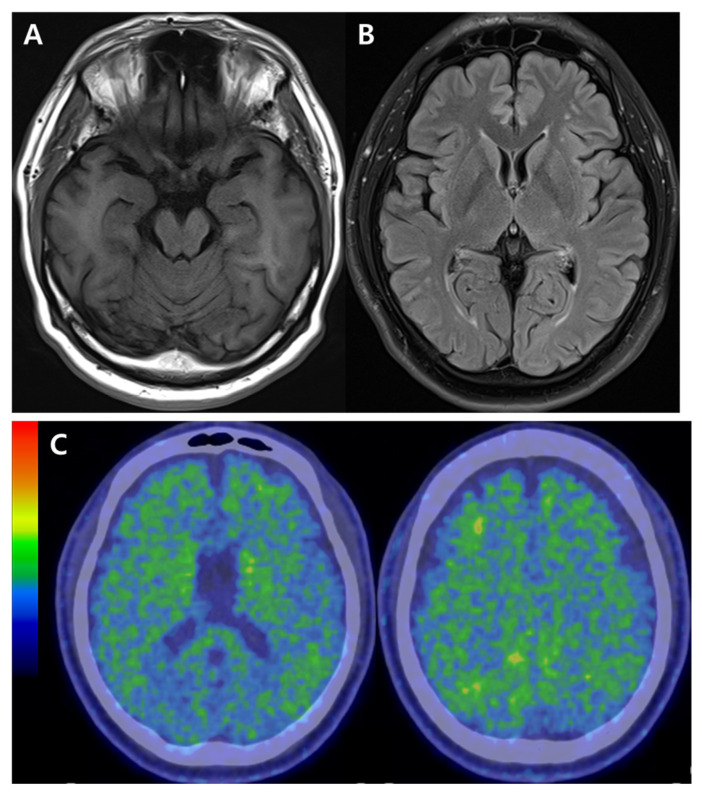
Neuroimaging data for the proband (III-1). (**A**). Magnetic resonance imaging (MRI) revealed mild hippocampal atrophy. (**B**). MRI showed a mild white matter change. (**C**). Amyloid-positron emission tomography (PET) indicated amyloid positivity in several brain regions (bilateral– lateral temporal, frontal, and parietal areas).

**Figure 3 ijms-25-00116-f003:**
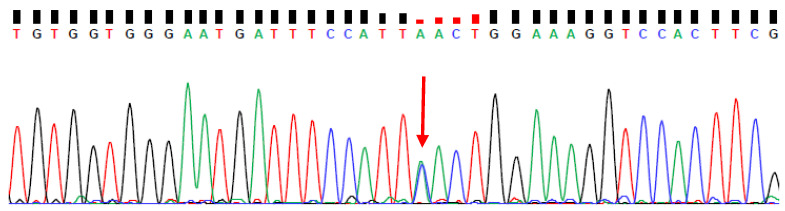
Sanger sequencing of patient with heterozygous *PSEN1* His214Asn. Red arrow shows the mutation.

**Figure 4 ijms-25-00116-f004:**
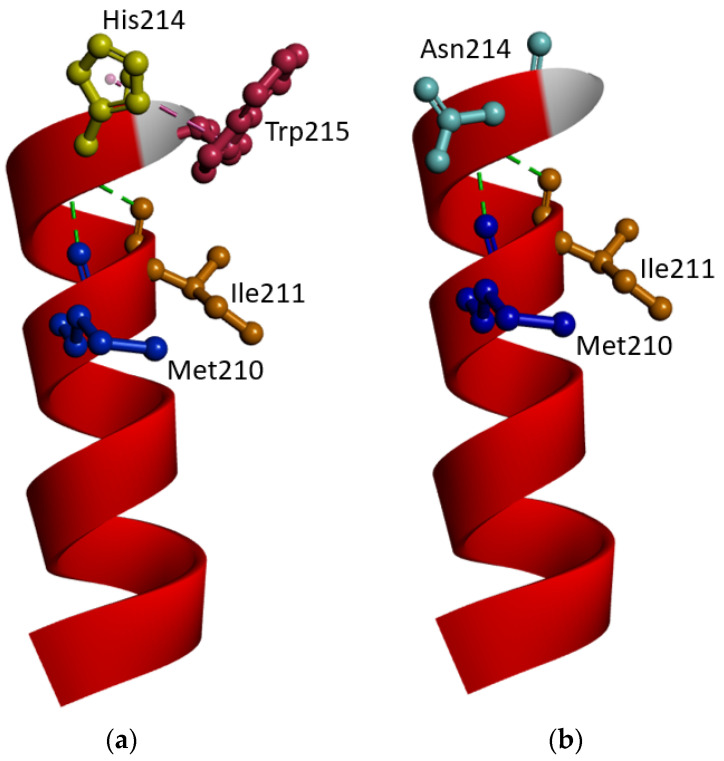
Structure predictions of (**a**) PSEN1 protein (helix-4) with His214 and (**b**) PSEN1 with Asn214. Even though the interactions between Asn214 to both Ile211 and Met210 remained in both structures, a cation-π interaction between Asn214 and Trp215 disappeared, suggesting the maintenance of helix within plasma membrane. The position of the helix would be affected, altering the γ-sec functions.

**Figure 5 ijms-25-00116-f005:**
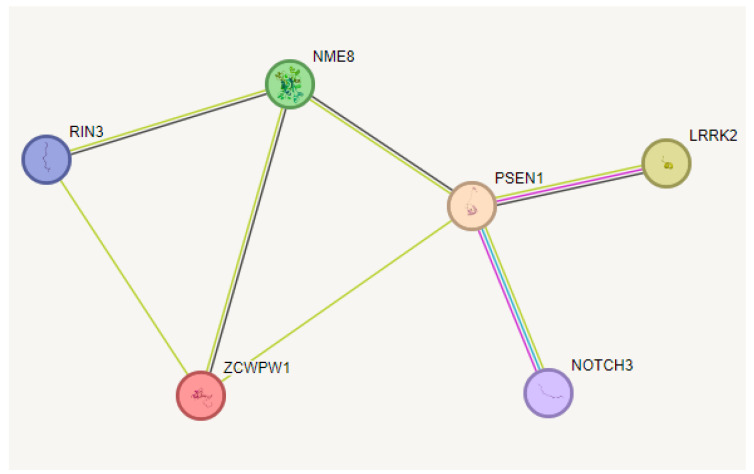
STRING analysis of rare variant-carrying genes found in the proband patient. Mutations in PSEN1 would affect its interactions with other correlating genes—LRRK2, NME8, NOTCH3, and ZCWPW1 directly and RIN3 indirectly. The green lines show that these proteins may interact based on “textmining”, and there is proof of interaction, based on literature. The black lines mean the genes may have evidence for co-expression. The blue line shows the gene interaction has been proven, based on curated databases (such as Biocarta or Reactome). The purple line shows the interaction had been proven experimentally. Length of lines are customizable.

**Figure 6 ijms-25-00116-f006:**
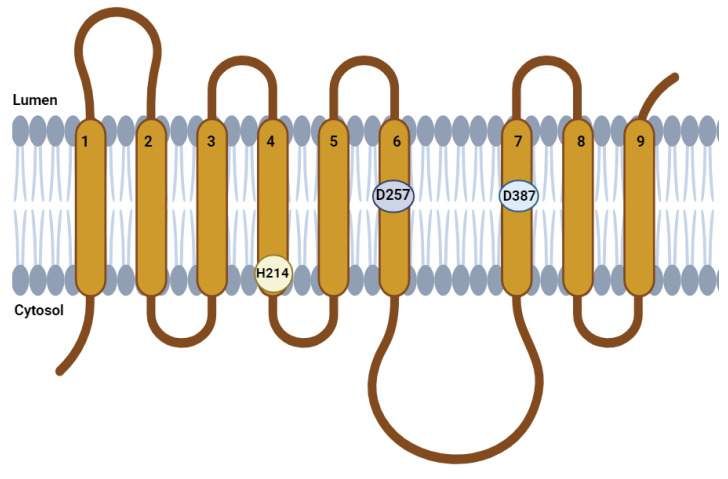
Location of His214 in the 4th transmembrane domain of PSEN1 protein. The position of the Asn214 mutation would affect the position of the helix in helix–helix interactions within plasma membrane and the dynamics of the γ-sec functions.

**Table 1 ijms-25-00116-t001:** Comparison of His214Asn, His214Asp, His214Tyr, and His214Arg mutations.

	His214Asn		His214Asp	His214Tyr		His214Arg
**Nationality of patients**	Korea	Italy	Spain	France	Iran	China
**Diagnosis**	EOAD	EOAD	Atypical EOAD	EOAD	EOAD, PD, FTD	EOAD
**AOO**	40s	40s–50s	55	37–45	50s	40s
**Family history**	Probably positive	Probably positive	Probably positive	Probably positive	Probably positive	Probably positive
**Symptoms**	Memory and executive dysfunctions, attention deficit, visuospatial dysfunction	Memory and executive dysfunctions, language impairment	Dementia with bradykinesia and tremor	Typical AD	Dementia, aphasia, motor impairment	Memory dysfunction, mood alterations, behavioral issues
**Biomarkers**	MDS-OAβ-positive	NA	CSF 14-3-3-negative	NA	NA	NA
**Imaging**	MRI: mild hippocampal atrophy FBB-PET: positive	CT: diffuse cerebral atrophy	NA	NA	MRI: global cortical and frontotemporal atrophy, white matter lesions	MRI: white matter demyelination in frontal and parietal lobes
***APOE* genotype**	33	33	33	NA	NA	33
**Functional data**	HEK293: reduced Aβ37/42 ratio, elevated Aβ42/40		HEK293: reduced Aβ37/42 ratio, elevated Aβ42/40	HEK293: reduced Aβ37/42 ratio, elevated Aβ42/40		NA
**Reference**	current case	[10]	[11]	[12]	[13]	[14]

Abbreviations: EOAD: early onset AD; PD: Parkinson’s disease; FTD: frontotemporal dementia; MDS: Multimer Detection System-Oligomeric Amyloid-β; CSF: cerebrospinal fluid; FBB-PET: ^18^F-Fluorbetaben Positron Emission Tomography; MRI: Magnetic Resonance Imaging; CT: Computed Tomography; NA: not available; APOE: apolipoprotein E, HEK293: human embryonic kidney 293 cells.

## Data Availability

Data available in a publicly accessible repository. Data is contained within the article and Appendix A.

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
