# Peer review of "PSEN1 His214Asn Mutation in a Korean Patient with Familial EOAD and the Importance of Histidine–Tryptophan Interactions in TM-4 Stability"

_ijms, 2023, doi:10.3390/ijms25010116_

Round 1

Reviewer 1 Report

Comments and Suggestions for Authors

IJMS-2769595. An interesting case requiring publication. But it requires corrections and explanations before publication. The manuscript requires editorial and linguistic corrections, especially in medical language. My comments, confusion and question. I think the word inclusions used by the authors is inappropriate. See lines 41, 43 and in other places of MS. Genes should be in italic. What it is line 52 and 53 (specific sign)? The sentence from lines 63-65 with data should be in methods. How was "whole blood" taken for heparin or clot? For reagents and equipment, please provide the company, city and country of origin.  Lines 86 and 90 is the word "presented", in my opinion, incorrect from a medical point of view. "Proband" line 99 and elsewhere in the manuscript is not the correct term. Has the patient's cerebral blood flow been assessed? Line 189 and elsewhere in MS what it is after “A”? Line 202 and 205 take out -. In table what it is MDS-A and something and the same is in line functional data. The abbreviations used in the table should be explained below the table. Line 228 what it is (2019)? Lines 236-237 something is missing in the sentence. Line 251 what it is before Func-? Line 262 editorial error. Line 263 what it is –sec? Literature should be presented in accordance with the editorial recommendations!

Comments on the Quality of English Language

Medical vocabulary should be checked and improved.

Author Response

Reviewer 1

Comments and Suggestions for Authors
IJMS-2769595. An interesting case requiring publication. But it requires corrections and explanations before publication.

Thank you very much, we appreciate the positive feedback and constructive comments.

The manuscript requires editorial and linguistic corrections, especially in medical language. My comments, confusion and question. I think the word inclusions used by the authors is inappropriate.

 See lines 41, 43 and in other places of MS. Genes should be in italic.

Thank you, we fixed this error.

 What it is line 52 and 53 (specific sign)?

Thank you, we fixed this error.

The sentence from lines 63-65 with data should be in methods.

Thank you, this error has been fixed.

 How was "whole blood" taken for heparin or clot?

Thank you, we fixed this error

Whole blood was drawn from proband patient with blood collection tube of EDTA anticoagulant (3.0 mL BD Vacutainer® plastic P700 plasma tube, NJ, USA). White blood cells were separated from plasma after the centrifugation on 800*g for 30 minutes.  The DNA was isolated from white blood cells with Qiagen Maxi blood kit (Seoul, Republic of Korea) by following the manufacturer’s protocol

 For reagents and equipment, please provide the company, city and country of origin. 

Thank you, we added the information.

 Lines 86 and 90 is the word "presented", in my opinion, incorrect from a medical point of view.

Thank you, we fixed this issue

 "Proband" line 99 and elsewhere in the manuscript is not the correct term.

Thank you, this issue has been fixed

“The genetic testing could not be performed on the above 3 members of the family due to their passing. All other living family members refused the genetic testing nor provided any further information, regarding their health (Figure 1).”

 Has the patient's cerebral blood flow been assessed?

Thank you, we fixed this issue.

“There were no significant steno-occlusive lesions or aneurysm in the intracranial artery in magnetic resonance angiography (MRA).’

 Line 189 and elsewhere in MS what it is after “A”?

Thank you, this error has been fixed. It is amyloid-beta or Ab.

Line 202 and 205 take out -.

Thank you, this error has been fixed.

In table what it is MDS-A and something and the same is in line functional data.

Thank you, this error has been fixed. It is amyloid beta or Ab.

 The abbreviations used in the table should be explained below the table.

Thank you, we added explanations of abbreviations.

Line 228 what it is (2019)?

. Cryoelectronic microscopy analysis by Zhou was published in 2019.

 Lines 236-237 something is missing in the sentence.

Thank you, this sentence has been rewritten

“In gas phase, histidine forms cation-π interactions with aromatic amino acids (such as Trp, Phe or Tyr), and among them the interaction between histidine and tryptophan was suggested to be the strongest. The energy of cation-π interaction between histidine and tryptophan could be -13.6 kcal/mole, which was higher than to the energy between histidine and phenylalanine or histidine between tyrosine (−7.809 and −7.887 kcal/mol, respectively).”

Line 251 what it is before Func-?

Gamma secretase. This error has been fixed

Line 262 editorial error. Line 263 what it is –sec?

Thank you, we fixed this error. It is gamma (γ)-secretase complex,

 Literature should be presented in accordance with the editorial recommendations!

Thank you, we tried to fix this issue.

Comments on the Quality of English Language

Thank you, we tried to improve the language.

Reviewer 2 Report

Comments and Suggestions for Authors

The presented manuscript describes the pathogenic PSEN1 His214Asn Mutation in a Korean Patient with Familial early-onset Alzheimer's disease (EOAD) and discusses the Importance of Histidine-Tryptophan Interaction in TM-4 Stability.

The entire manuscript is well-written and nicely illustrated (main text and supplementary data) and has a sufficiently explained Materials and Methods and Results section. The following discussion section is appropriate and complemented with a corresponding and up-to-date reference list.

However, minor details must be corrected before the manuscript is ready for publication.

These include:

Figure 1. Family tree of proband patient with His214Asn (III-1)- the short description of family member III-1 is missing (proband brother). Only three family members with a disease background were described, while four are presented in Figure 1.

The authors should also provide additional information about the STRING and Cluego tools (web page, software- Cytoscape plug-in for Cluego tool... ).

A minor revision is suggested.

Author Response

The presented manuscript describes the pathogenic PSEN1 His214Asn Mutation in a Korean Patient with Familial early-onset Alzheimer's disease (EOAD) and discusses the Importance of Histidine-Tryptophan Interaction in TM-4 Stability.

The entire manuscript is well-written and nicely illustrated (main text and supplementary data) and has a sufficiently explained Materials and Methods and Results section. The following discussion section is appropriate and complemented with a corresponding and up-to-date reference list.

However, minor details must be corrected before the manuscript is ready for publication.

Thank you very much, we really appreciated the positive feedback and constructive comments.

These include:

Figure 1. Family tree of proband patient with His214Asn (III-1)- the short description of family member III-1 is missing (proband brother). Only three family members with a disease background were described, while four are presented in Figure 1.

Thank you, we fixed the figure and family description.

The authors should also provide additional information about the STRING and Cluego tools (web page, software- Cytoscape plug-in for Cluego tool... ).

Thank you, this issue has been fixed:

For the association analyses between the different mutation carrier genes, STRING (https://string-db.org/, accessed on 1 October 2023) and Cytoscape Cluego tools were used version 3.10.0 (https://apps.cytoscape.org/apps/cluego, accessed on 1 October 2023 [7].